# High cognitive violation of expectations is compromised in cerebellar ataxia

Leonardo Daniel[1,2], Eli Vakil[3], William Saban[1,2]*

[1]Center for Accessible Neuropsychology and Sagol School of Neuroscience, Tel Aviv University, Tel aviv, Israel; [2]Department of Occupational Therapy, Gray Faculty of Medical & Health Sciences, Tel Aviv University, Tel Aviv, Israel; [3]Department of Psychology and Leslie and Susan Gonda (Goldschmied) Multidisciplinary Brain Research Center, Bar-Ilan University, Ramat-Gan, Israel

## eLife Assessment

This **valuable** investigation provides new and **solid** evidence for a specific cognitive deficit in cerebellar degeneration patients. The authors use three tasks that modulate complexity and violations of cognitive expectations. They show specific slowing of reaction times in the presence of violations but not with task complexity. While some alternative interpretations of the results are possible and are discussed, the work provides a new, invaluable data point in describing the cognitive contribution of cerebellar processing.

*For correspondence:
williamsaban@gmail.com

**Competing interest:** The authors declare that no competing interests exist.

**Abstract** While traditionally considered a motor structure, the cerebellum is also involved in cognition. However, the underlying cognitive mechanisms through which the cerebellum contributes to evolutionarily novel cognitive abilities remain poorly understood. Another open question is how this structure contributes to a core unifying mechanism across domains. Motivated by the evolutionary principle of neural reuse, we suggest that a successful account of cerebellar contributions to higher cognitive domains will build on the structure's established role in motor behaviors. We conducted a series of neuropsychological experiments, assessing selective impairments in participants with cerebellar ataxia (CA) compared to neurotypicals in solving sequential discrete problems. In three experiments, participants were asked to solve symbolic subtraction, alphabet letter transformation, and novel artificial grammar problems, which were expected or unexpected. The CA group exhibited a disproportionate cost when comparing unexpected problems to expected problems, suggesting that the cerebellum is critical for violation of expectations (VE) across tasks. The CA group impairment was not found either when the complexity of the problem increased or in conditions of uncertainty. Together, these results demonstrate a possible causal role for the human cerebellum in higher cognitive abilities. VE might be a unifying cerebellar-dependent mechanism across motor and cognitive domains.

## Introduction

Traditionally, the cerebellum has been primarily recognized as a motor structure. However, evidence indicates its involvement in various cognitive processes (*Strick et al., 2009*; *Saban et al., 2024*; *Middleton and Strick, 2000*; *Guell et al., 2018*; *Ravizza et al., 2006*; *Hull, 2020*). Yet, the underlying cognitive mechanisms through which the cerebellum contributes to higher cognitive abilities remain unknown. In addition, since the anatomy of the cerebellum is relatively uniform throughout its structure, it was suggested that its function may be consistent (*Guell et al., 2018*; *Stoodley et al., 2012*) (i.e. 'universal cerebellar transform'). This led to the hypothesis that a cerebellar lesion will result in

a core ubiquitous impairment across domains (*Guell et al., 2018*; *Stoodley et al., 2012*; *Stoodley, 2012*; *Schmahmann, 2004*). Despite these theoretical proposals, direct evidence for the cerebellum's contribution to a core unifying mechanism across non-motor domains is lacking.

A substantial body of neuropsychological, modeling, and imaging evidence supports the notion that the cerebellum is involved in motor control (*Hull, 2020*; *Stoodley et al., 2012*; *Heffley et al., 2018*; *Debas et al., 2010*; *Imamizu et al., 2000*; *Ito, 2008*; *Wolpert and Miall, 1996*). According to the forward model framework (*Ito, 2008*; *Wolpert and Miall, 1996*), the cerebellum encodes a predictive model (*Wolpert et al., 1998*) (i.e. 'internal model'). Specifically, the cerebellum is involved in prediction error (*Wolpert et al., 1998*), such that processing of the deviation between the predicted and the perceived information (i.e. error signal) leads to an internal model. Research shows that individuals with cerebellar pathology exhibit impairments in a range of sensorimotor tasks (*Saban and Ivry, 2021a*; *Taylor et al., 2014*). For example, in visuomotor rotation tasks, people with CA exhibit a reduced ability to process experimental perturbations, where the participant's predicted outcome is different from the perceived stimuli. This is consistent with the idea (*Hull, 2020*; *Fiez et al., 1992*; *Moberget et al., 2014*; *Sokolov et al., 2017*) that an intact cerebellum is required to process VE. However, it is not fully understood how the concept of VE should be applied in cognitive domains (*Ito, 2008*; *McDougle et al., 2022*).

The cerebellum is involved in many cognitive abilities, such as sequence learning, executive function, and even math and language processes (*Strick et al., 2009*; *Saban et al., 2024*; *Ravizza et al., 2006*; *Schmahmann, 2004*; *Sokolov et al., 2017*; *Lesage et al., 2016*; *Lesage et al., 2017*; *Saban and Gabay, 2023*; *Caligiore et al., 2017*; *Bostan and Strick, 2018*; *Barth et al., 2010*; *Nicolson et al., 2010*). This is evidenced by cognitive deficits in individuals with cerebellar pathology (*Saban et al., 2024*; *Saban and Ivry, 2021a*; *McDougle et al., 2022*), bidirectional connectivity with the neocortex (*Caligiore et al., 2017*; *Bostan and Strick, 2018*; *Watson et al., 2014*), developmental studies (*Riva and Giorgi, 2000*), and neuroimaging studies showing cerebellar activation in cognitive tasks (*Stoodley et al., 2012*; *Lesage et al., 2016*; *Walz et al., 2015*). While existing studies support a cerebellar role in nonmotor functions (*Strick et al., 2009*; *Ravizza et al., 2006*; *Fiez et al., 1992*; *Riva and Giorgi, 2000*; *Schmahmann, 2019*; *Buckner, 2013*; *Nicholas et al., 2024*), direct evidence and stronger theoretical constraints on the region's specific function in higher cognition are needed (*Saban and Gabay, 2023*). In addition, despite the broad evidence in many cognitive domains, how this structure contributes to a fundamental, unifying cognitive mechanism remains an open question (*Stoodley, 2012*; *Guell et al., 2017*).

Furthermore, we observed that imaging and neuropsychological literature do not always converge, with the latter mainly showing mixed results. For instance, one language paradigm that activates the cerebellum is semantic processing, focusing on right cerebellar activation (*Stoodley, 2012*; *Moberget et al., 2014*; *Lesage et al., 2017*; *Fiez et al., 1996*; *Murdoch, 2010*). However, while some studies did not find that patients with cerebellar pathology have impairments in semantic processing tasks (*Gasparini et al., 1999*; *Riva, 1998*; *King et al., 2024*), others found significant impairment (*Moberget et al., 2014*), specifically related to the processing of errors (*Fiez et al., 1992*) or sequential processing (*Riva, 1998*). Another example is working memory. The neuroimaging literature demonstrated the association of cerebellar activity with working memory tasks (*Stoodley et al., 2012*; *E et al., 2014*). However, again, the results are not consistent in the neuropsychology literature. While some studies report impairments in individuals with cerebellar pathology on working memory tasks (*Ziemus et al., 2007*), others report null effects (*McDougle et al., 2022*; *Appollonio et al., 1993*). These mixed results may reflect the heterogeneity in the cerebellar patient samples, or it may be that the chosen cognitive tasks are not consistently sensitive to cerebellar pathology. To address these two challenges, we proposed to examine a homogenous sample of individuals with a single type of spinocerebellar ataxia (SCA6). Additionally, we suggested utilizing tasks designed to be more reliably sensitive in capturing the specific cognitive impairments associated with cerebellar dysfunction.

Additionally, in the motor domain, models of cerebellar function emphasize the importance of this structure in VE (*Miall et al., 2007*; *Wolpert and Flanagan, 2001*). Expectations, which are based on top-down knowledge, are compared with visual feedback, with the difference between the two serving as an error signal (*Wolpert et al., 1998*). However, given that violation of expectations is a general feature of brain function (*Friston, 2018*), a key challenge is to specify constraints on the cerebellum's specific role. When we examined previously used tasks, we noticed that many of them

required procedure-based sequential processing (*Strick et al., 2009*; *Saban et al., 2024*; *Fiez et al., 1992*; *Moberget et al., 2014*; *Riva, 1998*; *Elyoseph et al., 2020*; *Morgan et al., 2021*; *Pascual-Leone et al., 1993*; *Molinari et al., 1997*; *Heleven et al., 2021*). People with CA show selective impairments in procedure-based tasks (*Saban et al., 2024*; *Saban and Ivry, 2021a*; *Elyoseph et al., 2020*; *Heleven et al., 2021*; *Knowlton et al., 2017*; *Knowlton et al., 2016*; *Adamaszek and Kirkby, 2016*), requiring an algorithm of a sequence of discrete steps to solve (*Saban and Ivry, 2021a*; *Riva, 1998*; *Elyoseph et al., 2020*; *Morgan et al., 2021*).

Herein, we proposed investigating the cognitive mechanisms by which the cerebellum contributes to tasks requiring sequential processing. Although defining the distinct cerebellum's role in cognition is challenging (*Saban and Gabay, 2023*; *Parvizi, 2009*), we anticipate that a successful theory will likely build on its well-established role in motor control (*Hull, 2020*; *Ito, 2008*; *Wolpert and Miall, 1996*; *Wolpert et al., 1998*; *Lesage et al., 2017*). Our logic is straightforward. VE is involved in almost every human function, from basic ones, such as motor control, to more complex ones, such as language and math. Evidence suggests that the cerebellum contributes to motor and cognitive tasks through the processing of VE (*Hull, 2020*; *Fiez et al., 1992*; *Elyoseph et al., 2020*; *Morgan et al., 2021*; *Molinari et al., 1997*; *Saywell and Taylor, 2008*). Motivated by the biological and evolutionary principle of neural reuse (*Rozin, 1976*; *Anderson, 2010*), we propose and test the hypothesis that the cerebellum also contributes to higher cognition through cognitive VE.

Evolutionarily novel cognitive abilities, such as arithmetic and language, also rely on the processing of VE (*Fiez et al., 1992*; *Moberget et al., 2014*; *Lesage et al., 2017*; *Berger et al., 2006*; *Argyropoulos, 2016*). For instance, based on known rules, a person predicts the correct answer to an equation, the next word in a sentence, or the next letter in a word. The expectations in these sequential processes are sometimes accurate and sometimes not. We suggest that through a VE mechanism, the cerebellum contributes not only to motor abilities but also to the cognitive procedures necessary for solving symbolic arithmetic and alphabet transformation problems (*Saban et al., 2024*; *Fiez et al., 1992*; *Lesage et al., 2017*).

Notably, these higher cognitive sequential problems are probably not solved by one continuous mental computation, and multi-step discrete computations are needed (*Dietrich and Markman, 2003*; *Logan and Klapp, 1991*; *Ashcraft, 1992*; *Verhaeghen et al., 1997*). It is more likely that mental arithmetic and alphabet transformation problems are solved using discrete mental procedures by breaking down these complex cognitive computations into smaller steps (*Dietrich and Markman, 2003*; *Verhaeghen et al., 1997*; *Cohen-Kadosh and Dowker, 2015*; *LeFevre et al., 1996*; *DeStefano and LeFevre, 2004*; *Grabner and De Smedt, 2011*; *Taatgen, 2013*; *Fias et al., 2021*; *Tiberghien et al., 2019*). For instance, when solving a simple subtraction problem (e.g. 9−5=3), a person typically follows a series of mental steps. For instance, first, they represent the stimuli mentally, understanding the need to subtract 5 from 9. They then retrieve relevant arithmetic facts from long-term memory or apply subtraction rules (*Cohen-Kadosh and Dowker, 2015*; *Grabner et al., 2009*). After arriving at a potential answer, they compare their expected computation (e.g. 4) with the actual visual information presented on the screen (e.g. 3). This algorithmic procedure highlights how sequential cognitive steps are used to solve even a simple arithmetic problem and how VE is a process needed for higher cognition as well.

Interestingly, in a sequential task requiring a series of discrete steps to solve, expectations are formed within each trial rather than between trials. During the operation of these mental steps, participants develop a prediction model (i.e. 'internal model'), predicting the correct outcome. When an incorrect answer is presented, it leads to a violation of the participant's expectation within that specific trial. This process allows for the investigation of VE within the context of a single trial, providing valuable insights into the underlying cognitive mechanism for each problem. Taken together, rather than being restricted to continuous transformations as suggested before (*McDougle et al., 2022*), we propose that the cerebellum is necessary for solving sequential higher cognitive problems via VE.

## Hypothesis and goals

Rather than merely implementing previously learned cognitive processes (e.g. memory retrieval *Saban et al., 2024*; *Elyoseph et al., 2020*), we propose that the cerebellum adapts procedures (i.e. 'internal model') by processing VE also in higher cognitive domains. The universal cerebellar transform theory (*Schmahmann et al., 2019*) suggests that since the cerebellum is a relatively uniform structure, its

function may also be uniform across domains (*Hull, 2020*; *Schmahmann, 2004*; *Lesage et al., 2017*; *Guell et al., 2017*). Thus, we hypothesized that cerebellar degeneration would result in a core ubiquitous impairment in processing VE across complex cognitive tasks.

Given the scarcity of literature in this field, which remains inconclusive (*Saban et al., 2024*; *Fiez et al., 1992*; *Lesage et al., 2017*; *Gasparini et al., 1999*; *King et al., 2024*; *Appollonio et al., 1993*; *Argyropoulos, 2016*), we first aimed to examine whether individuals with cerebellar degeneration are impaired in three high cognitive tasks, which require sequential processing. Second, we aim to assess the cerebellum's contribution, specifically in processing VE, compared to other cognitive processes (e.g. task complexity). Third, we asked whether the processing of VE could be a core, domain-general cerebellar-dependent mechanism across these cognitive tasks.

To investigate the cerebellum's role in processing error signals in high cognitive processes, we employed a neuropsychological approach, comparing the performance of individuals with CA to NT. Our study consisted of three experiments designed to measure participants' responses to correct and incorrect stimuli, focusing on the effect of errors on participants' behavior.

In Experiments 1 and 2, we aimed to assess whether CA modulates the processing of VE in two different tasks that require sequential processing: Subtraction reasoning and transformation of alphabet letters. Participants were required to identify whether a given problem was correct or incorrect. In each task, we manipulated two processes: the effect of errors on participants' responses and, for comparison, the effect of task complexity (e.g. the number of mental steps). Comparing these two effects allowed us to assess potential confounds, such as task difficulty and perceptual processes (*Saban et al., 2024*). Given our hypothesis and previous findings (*Moberget et al., 2014*), we predicted that the CA group would show a distinct disproportionate cost for VE (unexpected minus expected problems) but not for the complexity effect (high complexity minus low complexity problems).

While in Experiments 1 and 2, participants identify if a given problem is correct or not based on their previous knowledge, in Experiment 3, we probed novel cognitive expectations under uncertainty. We formed a new scenario where lifetime learned expectations are absent (no prior knowledge), and participants are required to learn a new artificial grammatical rule within the task. The participants' expectations were novel and not based on previous top-down processes, allowing for minimizing memory effects. To do so, we utilized an artificial grammar learning (AGL) task (*Chang and Knowlton, 2004*; *Pothos, 2007*). Given previous studies demonstrating the cerebellum's role specifically in procedure-based learning (*Saban et al., 2024*; *Ito, 2008*; *Elyoseph et al., 2020*; *Morgan et al., 2021*; *Doya, 2000*), we predicted that the cerebellum would contribute to procedure-based learning (*Fiez et al., 1992*; *Riva, 1998*) of an artificial grammar. To assess the effect of uncertainty on participants' accuracy, we manipulated the level of similarity between grammatical and nongrammatical problems. We predicted that the CA group would exhibit selective impairment only in the low similarity condition, where there is greater certainty, making it easier to differentiate between grammatical and nongrammatical problems.

Notably, given that processing VE is highly dependent on previous knowledge and top-down memory-related processes, it is important to consider these factors. Thus, across the three experiments, we examined VE in both established and newly learned cognitive procedures, considering varying levels of potential top-down effects. Furthermore, we aimed to determine whether the between-group differences in the cost of processing VE are domain-general in three cognitive tasks.

## Results

### Experiment 1 – Cerebellar contribution to symbolic subtraction

In typical paradigms assessing cerebellar contributions to motor processes (*Taylor et al., 2014*; *Tzvi et al., 2017*; *Bares et al., 2007*), deliberate incorrect information is displayed on a computer screen using visual perturbation. The incorrect information contradicts the expected (correct) visual input. This experimental manipulation aligns with the notion that an intact cerebellum is crucial in processing error signals. Similarly, an event-related potentials study by *Berger et al., 2006* by Posner and others utilized correct equations (e.g. 1 + 1 = 2) in which the presented solution (e.g. 2) matched participants' expectation. However, in incorrect equations (e.g. 1 + 1 = 1), the presented solution (e.g. 1) violated the expected results and creates a cognitive VE. Thus, to maintain a comparable experimental

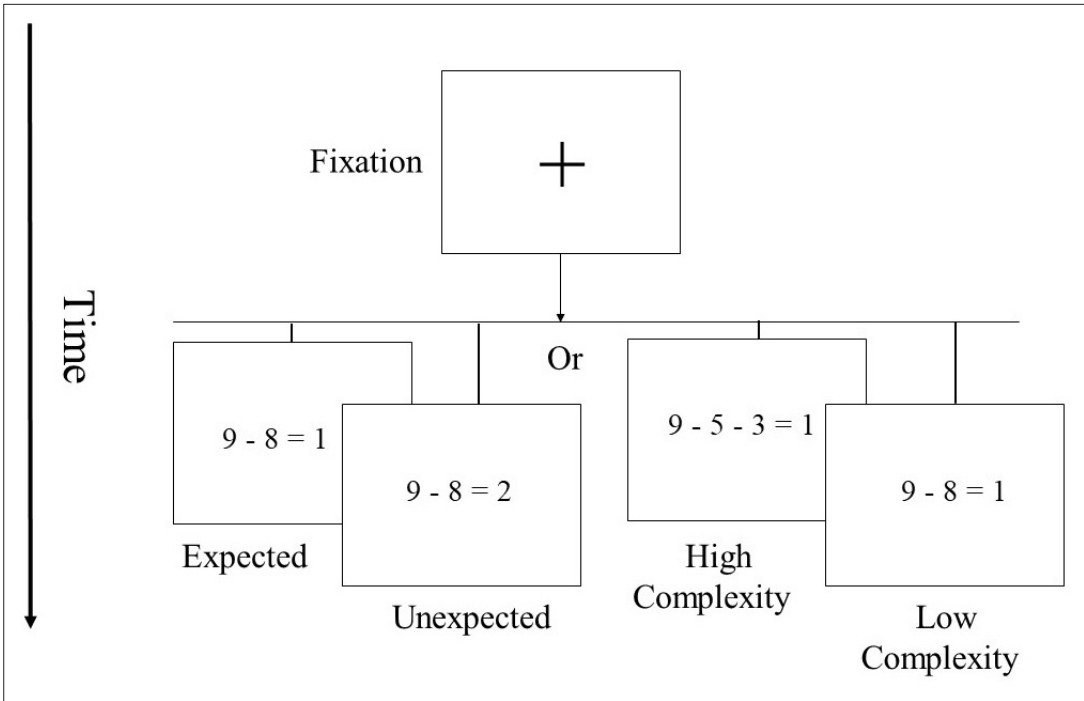

**Figure 1.** Experiment 1 – Subtraction verification task.

paradigm in the cognitive domain, while also having a baseline assessment, we compared correct and incorrect cognitive problems. Comparing incorrect to correct problems allowed us to control for potential confounds, such as memory and perceptual processes (*Moberget et al., 2014*; *Lesage et al., 2017*; *Cohen-Kadosh and Dowker, 2015*; *Ashcraft and Battaglia, 1978*).

In Experiment 1, participants completed a subtraction verification task (see *Figure 1*). We utilized a subtraction task because, compared to addition or multiplication that were used in previous studies (*Saban et al., 2024*; *McDougle et al., 2022*; *Flaumenhaft et al., 2025*), subtraction necessitates more spatial procedural processes (*Ward et al., 2009*; *Dormal et al., 2014*; *Díaz-Barriga Yáñez et al., 2020*), which are directly related to known cerebellar functions (*Mandolesi et al., 2003*; *Leggio et al., 2000*; *Mandolesi et al., 2001*). We used manipulation to probe two major subtraction processes. First, we manipulated the expectancy effect (expected vs. unexpected) of the equation. While at correct equations (e.g. 9–5–1=3), the presented solution (e.g. 3) is in accordance with the participant's expectation, at incorrect equations (e.g. 9–5–1=4), the presented solution (e.g. 4) violates the participant's expectations (see also *Berger et al., 2006*). Accordingly, response times (RT) are typically higher for incorrect problems compared to correct problems (*Ashcraft, 1992*; *Cohen-Kadosh and Dowker, 2015*; *Saban et al., 2021c*). We computed the 'expectancy effect' score by subtracting the mean RT of expected equations from the mean RT of unexpected equations. Second, we manipulated the complexity of the equation by probing the number of steps required to solve it. We employed problems that involved either one or two operators. Accordingly, subtracting one single-digit number (e.g. 9–5=4) takes less time than subtracting two single-digit numbers (9–3–2=4) because the latter requires more cognitive steps to solve the problem (*Saban et al., 2024*). We computed the 'complexity effect' score by subtracting the mean RT of low-complexity trials from that of high-complexity trials. RT was calculated on the participants' correct responses.

*Figure 2* shows RTs as a function of group (NT/CA), expectancy (expected/unexpected), and complexity (high/low). We utilized a linear mixed-effects (LME) (*Bates et al., 2015*; *Figner et al., 2020*) model with group, expectancy, and complexity as fixed effects, and participant ID as a random factor.

To establish a baseline, we assessed the NT group's performance. As predicted, the NT groups showed a significant expectancy effect (beta for simple effect estimator (est.)=238 ms, $p<0.0001$) and a significant complexity effect (est.=831 ms, $p<0.0001$). The CA group exhibited

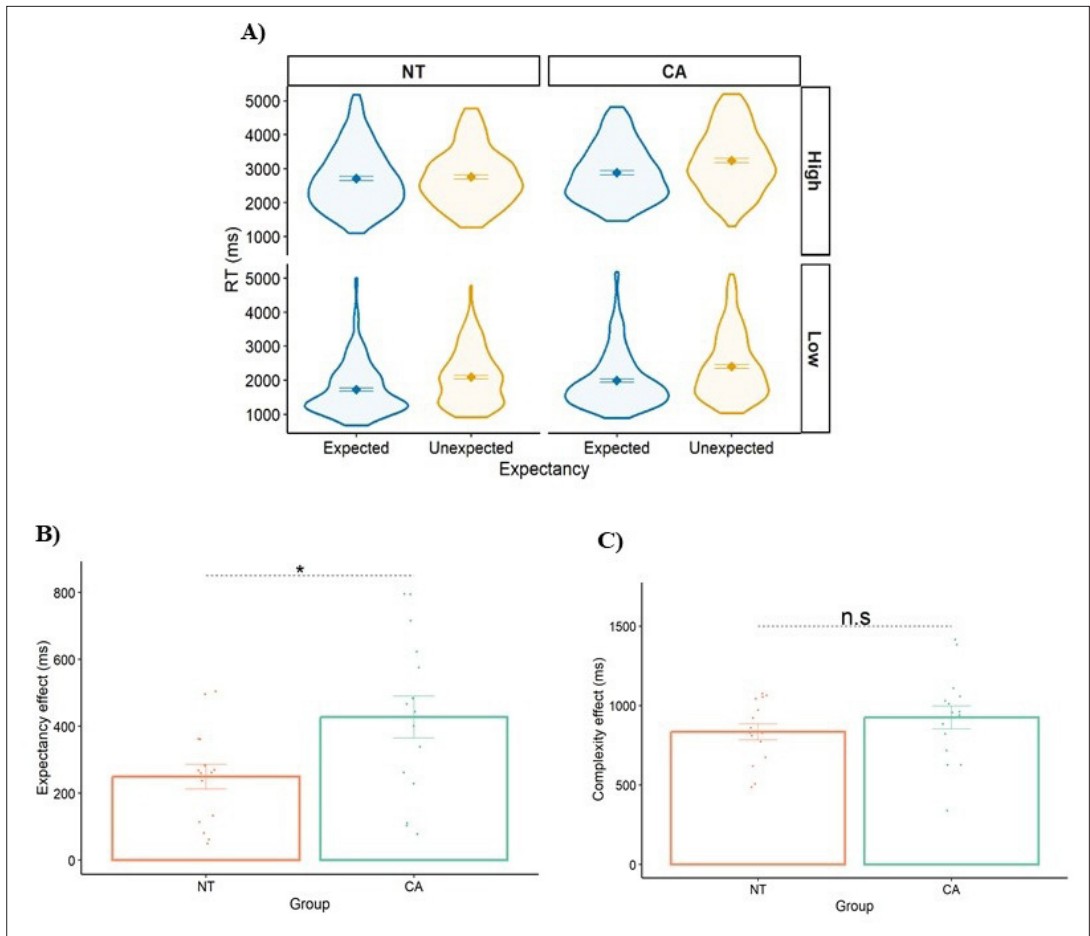

**Figure 2.** Results of Experiment 1. (**A**) Response times (RTs) as a function of group (neurotypical. NT/cerebellar ataxia, CA), expectancy (unexpected/expected), and complexity (high/low). (**B**) The expectancy effect (RT(unexpected) minus RT(expected)) for each group. (**C**) The complexity effect (RT(high) minus RT(low)) for each group. Each data point is a participant. Error bars = SEM. * indicates *p*=0.009. n.s=not significant.

a quantitatively slower response than the NT group across all conditions, but this difference did not reach statistical significance (main effect est.=304 ms, *p*=0.120). Our focus in Experiment 1 is comparing the effects of expectancy and complexity between the CA and NT groups (*Figure 2B*). Notably, the expectancy effect was significantly larger for the CA group compared to the NT group (two-way interaction: est.=160 ms, *p*=0.013, Cohen's d=0.897; large effect size), but there was no significant difference between the groups in the complexity effect (two-way interaction: est.=79 ms, *p*=0.229).

In terms of covariates, there were no significant differences between the groups in years of education and age (*p*>0.05). In addition, accuracy rates were not significantly different between the groups (NT = 91%, CA = 88%, *p*=0.297). Furthermore, there was no significant interaction between Group and Expectancy when accuracy was the dependent variable (est.=3.5, *Pp*=0.61). This similar accuracy rates indicate that accuracy was not a sensitive measure for detecting group differences in this task.

To conclude, two primary insights can be derived from the results of Experiment 1. First, the CA group demonstrated a deficiency in a symbolic subtraction task, providing novel evidence of the cerebellum's contribution to higher cognitive functions. Second, the findings indicate a selective impairment of the CA group compared to the NT group. The CA group exhibited a disproportionate expectancy effect but not a complexity effect. In line with the established role of the cerebellum in the motor domain, we propose that the selective impairment reflects the cerebellum's role in processing arithmetic VE.

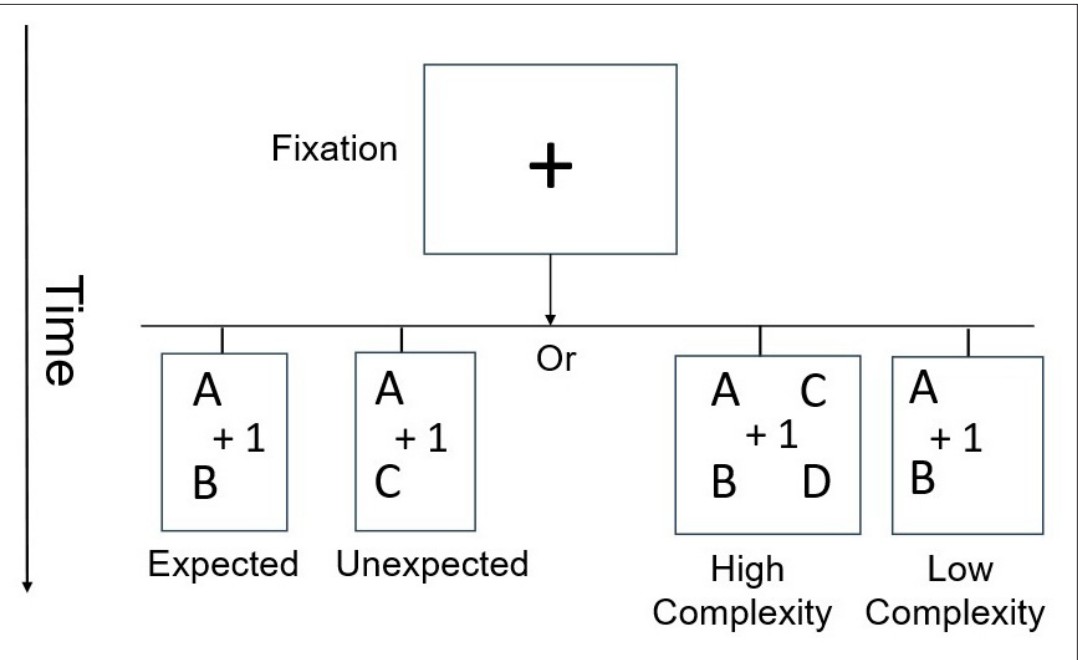

**Figure 3.** Experiment 2 – Alphabet transformation task.

## Experiment 2 – Cerebellar contribution to alphabet transformation

Participants completed an alphabet letter sequential transformation task with similar design to the previous task (see *Figure 3*). However, the design of this task was specifically tailored to achieve the following two objectives. First, we wanted to generalize the findings from arithmetic digits to alphabet letters. Participants were required to determine the correctness of an alphabet letter rule-based transformation. Similar to Experiment 1, the transformation rule was an arithmetic operator (e.g. +1), but the stimuli were alphabetic letters (e.g. F, K, L). This required participants to utilize their understanding of alphabetical sequences and relationships for each step in a discrete controlled manner. Second, we aimed to reduce top-down potential effects. While most educated participants have experience solving arithmetic equations, they probably have less experience transforming alphabetic letters using an arithmetic operator. Consequently, the alphabet transformation task reduces reliance on prior numerical proficiency. This allows for assessing VE in the context of relatively newly formed procedures. Participants were required to determine the correctness of the transformation displayed on the screen.

*Figure 4* shows RTs as a function of group (NT/CA), expectancy (expected/unexpected), and complexity (high/low). We utilized a linear mixed-effects (LME) (*Bates et al., 2015*; *Figner et al., 2020*) model with group, expectancy, and complexity as fixed effects, and participant ID as a random factor.

As predicted, participants responded slower to the unexpected than the expected alphabet transformation problems (the expectancy effect, est.=327 ms, $p<0.0001$). In addition, participants responded slower to the high complexity condition (two-letter problems) compared to the low complexity condition (the complexity effect, est.=1129 ms, $p<0.0001$). Across all trials, we observed that the CA group was significantly slower than the NT group (est.=1470 ms, $p<0.0001$).

Our focus in Experiment 2 is on the comparison of the effects of expectancy and complexity between groups (*Figure 4B*). Notably, only the expectancy effect was significantly larger for the CA group compared to the NT group (expectancy effect: est.=372 ms, $p<0.0001$, Cohen's $d$=1.388 (large); complexity effect: est.=108 ms, $p$=0.270). In terms of covariates, there were no significant differences between the groups in years of education, MoCA, and age ($p>0.05$). In addition, accuracy rates were not significantly different between the groups (NT = 75%, CA = 76%, $p>0.05$). Furthermore, there was no significant interaction between group and expectancy when accuracy was the dependent variable

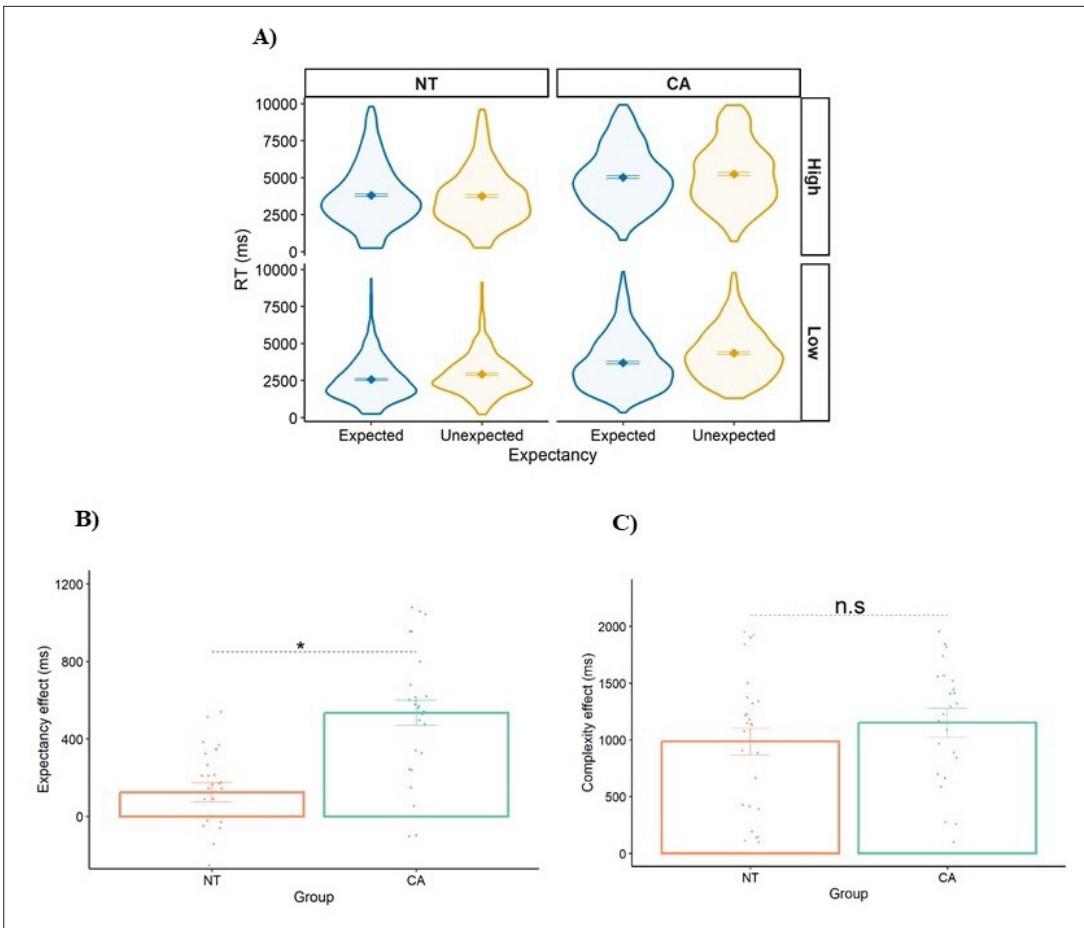

**Figure 4.** Results of Experiment 2. (**A**) Response times (RTs) as a function of group (neurotypical, NT/cerebellar ataxia, CA), expectancy (unexpected/expected), and complexity (high/low). (**B**) The expectancy effect (RT(unexpected) minus T(expected)) for each group. (**C**) The complexity effect (RT(high) minus RT(low)) for each group. Each datapoint is a participant. Error bars = SEM. * indicates $p<0.0001$. n.s=not significant.

(est.=4.0, $p$=0.277). These similar accuracy rates indicate that accuracy is not a sensitive measure for detecting group differences in this experiment.

Consistent with Experiment 1, in Experiment 2, using the alphabet transformation task, the CA group showed a selective disproportionate expectancy effect, but not complexity effect. In Experiment 2, the required transformation was implemented on alphabetic letters, which allowed for the generalization of the results from arithmetic digits to language-related stimuli. Furthermore, transforming alphabetic letters by an arithmetic operator is a less commonly used cognitive operation than subtracting digits, as evidenced by the general increase in RT (~5000 ms vs. ~3000 ms). This allowed us to reduce the potential effects of top-down processes and show that CA distinct impairment also appears in these conditions.

## Experiment 3 – Cerebellar contribution to cognitive expectations under uncertainty

In Experiment 3, we aimed to test our hypothesis in novel scenarios involving cognitive expectations under uncertainty. We manipulated the participants' expectations by forming new predictions – learning new rules (i.e. grammar) without prior knowledge of the required procedures. Participants were first exposed to strings of letters generated according to a specific novel grammatical rule. Next, they were shown new strings and asked to judge whether these strings followed the grammatical rules or not. We assessed learning by measuring how well participants could distinguish between grammatical and nongrammatical strings.

During the test phase, we assessed the participant's response to unexpected (nongrammatical) versus expected (grammatical) problems. As there are no expectations based on previous lifetime knowledge, the participant's expectations are formed during the task rather than before it. This approach allowed us to minimize the influence of previously learned top-down processes on newly learned grammar regarding a specific order of a sequence of letters. We achieved this by using an AGL task (*Chang and Knowlton, 2004*).

Notably, we also probed novel cognitive expectations under uncertainty. In the test phase, to assess the effect of uncertainty, we manipulated the level of similarity between grammatical and nongrammatical strings. In the low similarity condition, it was relatively easy for participants to distinguish between grammatical and nongrammatical strings. Participants were able, with a higher level of sensitivity, to differentiate which problems adhere to the expected grammar and which deviate from it (i.e. incorrect). In contrast, in the high similarity condition, participants' sensitivity was decreased. The discrimination between grammatical vs. nongrammatical strings was more challenging given the elevated degree of similarity between them. This manipulation also allowed us to disentangle between the difficulty in learning the rule during training and the difficulty in responding to the new strings during testing. Similar to previous studies (*Chang and Knowlton, 2004*), we then categorized the test strings into four groups based on grammaticality and low- or high-similarity values: grammatical high, grammatical low, nongrammatical high, and nongrammatical low.

To describe the magnitude of the sensitivity to grammatical status, we calculated a percent correct score for each participant in each condition. The analysis of accuracy not only provided insights into cognitive performance but also served a secondary benefit by reducing potential motor effects on RT. This approach helps to disentangle cognitive processes from motor execution, which is impacted by CA.

Typically, in experiments using the AGL task (*Chang and Knowlton, 2004*), the NT group is more accurate in grammatical strings compared to nongrammatical strings. In addition, this group demonstrates higher accuracy in low similarity conditions, where grammatical strings are likely more distinguishable from non-grammatical strings, compared to high similarity conditions. Accordingly, and to control potential response bias/motor-related abilities, a common dependent measure is $d'$ (sensitivity) in discriminating between grammatical vs. nongrammatical strings. To investigate the expectancy effect under uncertainty conditions, our focus in Experiment 3 was on the comparison of the $d'$ between groups in each level of similarity. Notably, when looking at $d'$, we predicted an interaction between group and similarity, such that the $d'$ will be larger for the NT group compared to the CA group in the low similarity condition only.

*Figure 5A* shows accuracy as a function of group (NT/CA), expectancy (expected/unexpected), and similarity (high/low). We utilized an LME model with group, expectancy, and similarity as fixed effects, and participant ID as a random factor.

We observed no differences between the groups in mean accuracy across conditions (57.4%, est.=4.02%, $p$=0.310), indicating that potential specific group differences in this task cannot be fully explained by motor abilities. Then, turning to our main variables of interest, as predicted (*Berger et al., 2006*), the NT group was more accurate in grammatical strings compared to nongrammatical strings (error effect est.=28.20%, $p$<0.0001). In addition, the NT group was more accurate in the low similarity condition compared to the high similarity condition (est.=11.33%, $p$=0.024). Accordingly, and to control for potential response bias, we also found higher $d'$ (sensitivity) in discriminating between grammatical vs. nongrammatical strings for the low similarity condition compared to the high similarity condition (est.=0.635, $p$<0.0001).

Our focus in Experiment 3 is on the comparison of the $d'$ between groups in each level of similarity (*Figure 5B*). Importantly, when looking at $d'$, we found an interaction between group and similarity (est.=0.459, $p$=0.022). As predicted, the $d'$ was larger for the NT group compared to the CA group in the low similarity condition only (est.=0.464, $p$<0.0001, effect size = 1.106 (large); High similarity: est.=0.004, $p$=0.978). Yet, one should notice that the absence of a group difference in the high-similarity condition could be attributed to a floor effect.

In terms of covariates, there were no significant differences between the groups in years of education, MoCA, and age ($p$>0.05). As expected, no significant group differences in RT were observed for this task ($p$=0.549, NT = 2568 ms, CA = 2815 ms). Additionally, the interaction between group and expectancy was insignificant when RT was the dependent variable (Low similarity: est.=255, $p$=0.310;

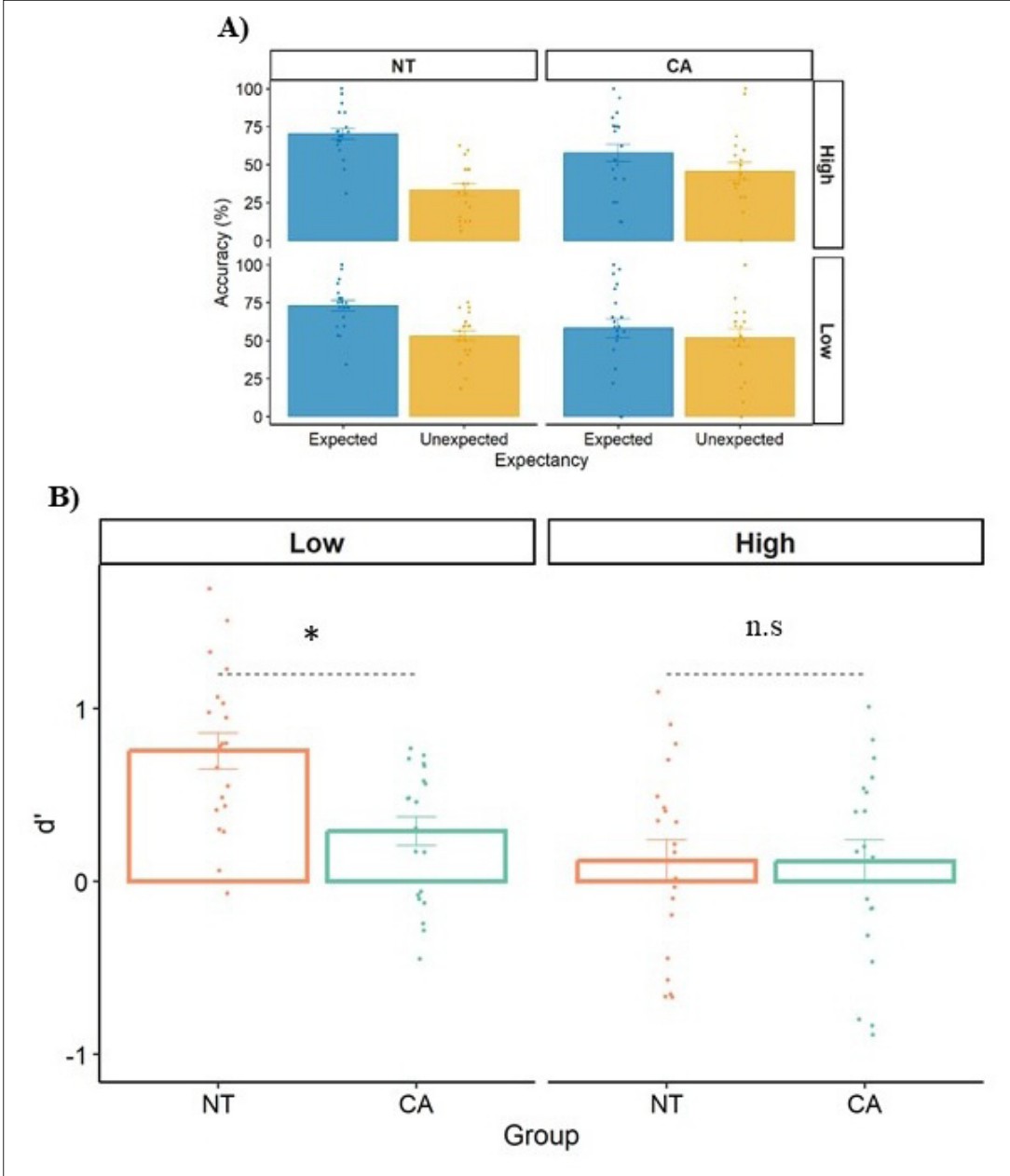

**Figure 5.** Results of Experiment 3. (**A**) Accuracy as a function of group (neurotypical, NT/cerebellar ataxia, CA), expectancy (expected/unexpected), and similarity (high/low). (**B**) Sensitivity (d') as a function of group for each level of similarity (high/low). Each datapoint is a participant. Error bars = SEM. * indicates $p<0.0001$. n.s=not significant.

High similarity: est.=352, $Pp$=0.180). Therefore, as commonly used in AGL tasks, we focused on accuracy measures.

We also conducted a criterion (c') analysis to examine potential effects of response bias. First, we did not find group differences in response bias (est.=0.291, $p$=0.213). Second, while we observed a two-way interaction between Similarity and Group when d' was the dependent variable, we did not find a significant interaction when we examined response bias (est.=0.114, $p$=0.262). This pattern of results supports the interpretation of reduced sensitivity in the CA group rather than the presence of a systematic response bias. Furthermore, we are unaware of any robust prior studies demonstrating a consistent response bias in the CA group.

Additionally, if the CA group had a bias to respond 'no' or exhibited any other bias, such a bias should have been evident across different conditions. However, our inclusion of a control condition

allowed us to examine this possibility. When we examined the High similarity condition, we did not find significant differences between the groups in d'. If the CA group exhibited a bias, one would expect significant performance differences from the control NT group across conditions and not only in the Low similarity condition. This alternative explanation, however, is not supported by the current empirical data.

## Discussion

In three experiments, patients with CA and neurotypical (NT) healthy participants solved symbolic arithmetic, alphabet transformation, and grammar sequential problems. These sequential problems required a series of discrete steps to be completed in a specific order to reach a solution. In Experiment 1, the CA group demonstrated impairment in a subtraction task, providing novel evidence of the cerebellum's contribution to symbolic arithmetic reasoning. In addition, we found a selective impairment of the CA group compared to the NT group. The CA group exhibited a disproportionate expectancy effect only (no difference in the complexity effect). Consistent with Experiment 1, in Experiment 2, using the alphabet transformation task, we found a distinct disproportionate expectancy effect (no difference in the complexity effect). Transforming alphabetic letters by an arithmetic operator is a less commonly used cognitive operation than subtracting digits. This allowed us to reduce the potential effects of top-down processes and show that CA's impairment also appears when there are fewer top-down effects. Rather, this probably increased the need to develop algorithmic procedures during the task.

In Experiments 1 and 2, participants identified if a given problem was correct or not based on their previous knowledge; however, in Experiment 3, we probed novel cognitive VE under uncertainty. We formed a new scenario where prior lifetime knowledge is absent, and participants are required to learn a new grammatical rule within the task. The participants' expectations were novel and were not based on previous top-down processes. Utilizing an AGL task *Pothos, 2007*, participants needed to learn a new Markovian grammar, regarding how to organize a sequence of letters in a specific order. To assess the effect of the participant's uncertainty, we manipulated the level of similarity between grammatical and nongrammatical problems. We found that the CA group showed selective impairment only in the low similarity condition (higher sensitivity), probably where expectations are higher. Although it was not a causal finding and no behavioral evidence was found, an fMRI study (*Lesage et al., 2017Lesage et al., 2017*) revealed that activity in the right posterolateral cerebellum correlated with the predictability of the upcoming target word. Together with our current findings and others (*Fiez et al., 1992*; *Moberget et al., 2014*), this pattern of results might indicate that the cerebellum is necessary for higher cognition through a VE mechanism.

Across the three experiments, we examined the effect of VE in both established and newly learned cognitive procedures, considering varying levels of potential top-down effects. The CA group showed a disproportionate expectancy effect compared to the NT group. We found that the between-group differences in the expectancy effect are consistent across tasks. Notably, the results indicate that CA patients had both intact processing of the problems' complexity (i.e. number of steps) and intact ability to discriminate between correct and incorrect problems when certainty decreased (i.e. sensitivity was lower). Thus, the results indicate a distinct role of the cerebellum in processing VE across these sequential cognitive tasks.

Several theories are in line with the principle of neural reuse and our hypothesis that the cerebellum contributes to many domains using the same core cognitive mechanism (*Saban and Gabay, 2023*). *Rozin, 1976* proposed that computations that initially evolved to solve specific problems become accessible to other systems through evolution, as well as within the individual lifetime of an organism. Change or expansion of a function, because it is more generally available or accessible, '*would have adaptive value when an area of behavioral function could profit from programs initially developed for another purpose.*' This idea has been reframed and elaborated upon in Gallese's 'neural exploitation' hypothesis (*Gallese and Cuccio, 2018*) and Anderson's 'massive redeployment' hypothesis (*Anderson, 2010*; *Anderson, 2007*). The core idea is that neural networks can acquire new uses after establishing an initial function. Decades of empirical research from human and animal experiments, including our own, support this framework (*Hull, 2020*; *Saban and Gabay, 2023*; *Saban et al., 2021b*; *Balsters et al., 2013*). The broad involvement of the cerebellum in motor and nonmotor functions

supports the idea of neural reuse, indicating this specific structure's potential ability to reuse its core function.

A few hypotheses have been proposed based on the idea that cerebellar contributions to motor control may extend to the cognitive domain (*Ito, 2008*; *Fiez et al., 1992*; *McDougle et al., 2022*). For example, in one recent paper (*McDougle et al., 2022*) it was hypothesized that the cerebellum supports dynamic continuous transformations of mental representations. However, the concrete implementation of 'continuity' in terms of higher mental representations remains an open question (*Dietrich and Markman, 2003*). Currently, there is no established direct evidence supporting the existence of such 'continuous' and 'dynamic' higher cognitive processes. Indeed, a recent later study by the same authors found no support for this hypothesis within the language domain (i.e. semantic processing task) (*King et al., 2024*). It is unclear whether the utilized problems, such as simple addition or mental rotation, are entirely solved using 'continuous transformation' or through previously learned procedural knowledge of the required mental steps. The constraints on the cerebellum's role remain an open question, particularly whether its role is limited solely to tasks requiring 'continuous mental transformation,' as previously suggested (*McDougle et al., 2022*). Therefore, more consistent and well-established cerebellar parsimonious theories are still needed.

One might be concerned that if the cerebellum is involved in sequential operations, its involvement in mental letter rotation, which can be assumed as 'continuous transformation,' may appear contradictory. We note that the boundary between continuous and stepwise, procedural operations is not always clear-cut and may vary depending on the participant's strategy and previous knowledge, which is not always fully known to the researchers. But this is a debatable consideration. More importantly, a careful reading of our paper suggests that our experiments were designed to examine VE within tasks that involve sequential processing. Notably, we are not claiming that the cerebellum is involved in sequential processing per se. Rather, our findings point to a more specific role for the cerebellum in processing VE that arises during the construction of multi-step procedural tasks. In fact, the results indicate that while the cerebellum may not be directly involved in the procedural process itself, it is critical when expectations are violated within such a context. This distinction is made possible in our study by the inclusion of a control condition (the complexity effect), which allows for a unique dissociation in our experimental design—one that, to our knowledge, has not been sufficiently addressed in previous studies (*McDougle et al., 2022*).

Additionally, in the case of arithmetic problem solving—such as the tasks used in prior studies (*McDougle et al., 2022*)—there is substantial evidence that these problems are typically solved through stepwise, procedural operations. Arithmetic reasoning, used in our Experiments 1 and 2, has been robustly associated with procedural, multi-step strategies, which may be more clearly aligned with traditional views of cerebellar involvement in sequential operations. Thus, we propose that the role of the cerebellum in continuous transformations should be further examined.

We suggest a more parsimonious theory – the cerebellum contributes to VE, a field that was highly examined before. Yet, to reconcile these findings, we propose that the cerebellum's contribution may not be limited to either continuous or stepwise, procedural operations per se, but rather to a domain-general process: the processing of VE. This theoretical framework can explain performance patterns across mental rotation tasks, grammar learning, and procedural arithmetic.

Another concern is related to oculomotor deficits (e.g. downbeat nystagmus), which are common in CA and could have influenced participants' performance. While the SARA scale does not assess oculomotor function, our experimental design – in all three experiments – has control conditions that help account for general processing differences, including those that could arise from oculomotor deficits. These conditions, such as the correct trials and the complexity effects, allow us to isolate effects specifically related to VE while minimizing the influence of broader performance factors, such as eye movement abnormalities. We also note that, while some patients can experience oculomotor symptoms such as downbeat nystagmus, none of our tasks required precise visual tracking or gaze shifts. In our experimental tasks, stimuli were centrally presented, and no visual tracking or saccadic responses were required.

Our study is subject to two main limitations. First, information about anatomical-behavioral relationships can provide valuable data and will certainly be important in the long run for understanding how the cerebellum contributes to cognition. For example, one can ask if performance is related to gross measures such as total cerebellar volume or finer measures such as whether the observed

deficits are associated with atrophy in particular regions. These analyses typically require large sample sizes, especially when dealing with atrophic processes (where the pathology tends to be relatively diffuse). Relatedly, we could not perform brain connectivity analysis, which limits our ability to examine the interactions between the cerebellum and other brain regions, such as the basal ganglia (BG) and the frontal lobe. This lack of brain connectivity data also restricts our ability to compare the cerebellum's functional contributions to those of other regions. For example, in a recently published paper (2024), we found that the BG plays a distinct role in mathematical complexity processes (*Saban et al., 2024*). Unfortunately, we do not have imaging data for many of the patients.

Yet, there is added value in studies that include behavioral results from neurological groups defined based on clinical diagnosis. This is common in the literature, as seen in other recently published studies (*Saban et al., 2024*; *McDougle et al., 2022*) in high-impact journals (PNAS, Journal of Neuroscience, Brain). In addition, the current experimental design is already complex, with two groups and three experiments, including control conditions. However, future work should use lesion analysis or connectivity methods to identify the specific anatomical-behavioral relationships critical for arithmetic and language operations.

A few important questions remain open in the literature concerning the cerebellum's role in expectation-related processes. The first is whether the cerebellum contributes to the formation of expectations or the processing of their violations. In Experiments 1 and 2, the CA group did not show impairments in the complexity manipulation. Solving these problems requires the formation of expectations during the reasoning process. Given the intact performance of the CA group, these results suggest that they are not impaired in forming expectations. However, in both Experiments 1 and 2, patients exhibited selective impairments in solving incorrect problems compared to correct problems. Since expectation formation is required in both conditions, but only incorrect problems involve a VE, we hypothesize that the cerebellum is involved in VE processes. We suggest that the CA group can form expectations in familiar tasks, but are impaired in processing unexpected compared to expected outcomes. This supports the notion that the cerebellum contributes to VE, rather than to forming expectations.

In Experiment 3, during training, participants learn a novel rule (grammar), forming new expectations on how strings of letters should be. Afterwards, during testing, the participants are requested to identify if a novel string following the rule or not. We examined sensitivity to distinguish between grammatical and non-grammatical strings of letters, thus taking into account a baseline ability to identify expected strings. Additionally, both in the low-similarity and high-similarity conditions, there are expectations regarding whether the strings following the rule or not. However, in the high-similarity condition, there is more uncertainty regarding which strings are following the grammatical rule, as demonstrated in a lower sensitivity (d prime). Given the group differences only in the low similarity condition, these results suggest the CA group is impaired only when the rules are more certain. Given these results, we suggest that forming cognitive expectations is not necessarily dependent on the cerebellum. Rather, we propose that the cerebellum is critical for processing VE (detection or processing of detected errors) under conditions of more certainty. One remaining question for future studies is whether the cerebellum contributes to detection of a mismatch between the expectation and sensory evidence, or the processing of a detected VE.

We suggest that these key questions are relevant to both motor and non-motor domains and were not fully addressed even in the previous, well-studied motor domain. Importantly, while previous experimental manipulations *Taylor et al., 2014*; *Moberget et al., 2014*; *Riva, 1998*; *Butcher et al., 2017* have provided valuable insights regarding the cerebellar role in these processes, some may have confounded these two internal constructs due to task design limitations (e.g. lack of baseline conditions). Notably, some of these previous studies did not include control conditions, such as correct trials, where there was no VE. In addition, other studies did not include a control measure (e.g. complexity effect), which limits their ability to infer the specific cerebellar role in expectation manipulation.

Thus, the current experimental design used in three different experiments provides a valuable novel experimental perspective, allowing us to distinguish between some, but not all, of the processes involved in the formation of expectations and their violations. For instance, to our knowledge, this is the first study to demonstrate a selective impairment in rule-based VE processing in cerebellar patients across both numerical reasoning and artificial grammar tasks. If feasible, we propose that future studies should disentangle different forms of VE by operationalizing them in experimental tasks

in an orthogonal manner. This will allow us to achieve a more detailed and well-defined cerebellar motor and non-motor mechanistic account.

## Conclusion

Our findings support neural reuse hypotheses (*Rozin, 1976*; *Anderson, 2007*) in that the cerebellum contributes to both motor and non-motor functions using a similar mechanism (*Hull, 2020*; *Moberget et al., 2014*; *Flaumenhaft et al., 2025*). The cerebellum not only contributes to motor control but also to cognitive procedures necessary for processing sequential arithmetic reasoning, alphabet transformation, and grammar problems using a similar mechanism – processing VE.

To conclude, theories of universal cerebellar transform propose that the cerebellum plays an essential role in modulating not only motor functions but also cognitive and affective processes (*Guell et al., 2018*; *Schmahmann, 2004*). The dysmetria of thought theory posits that cognitive and affective symptoms observed in cerebellar patients arise from the same dysfunction that affects motor control. This concept highlights the interconnectedness of motor, emotional, and cognitive domains, suggesting that impairments in one domain can reflect core problems in others. Our study provides convergent empirical evidence for the potential core role of the cerebellum and aligns with these previous theoretical frameworks.

## Methods

### Participants

The sample size was determined based on previous comparable work in CA patients. To calculate the required sample size, we conducted a power analysis (alpha = 0.05; power = 0.95) using effect sizes (Cohen's d=1.36) derived from previous studies (*Saban et al., 2024*; *McDougle et al., 2022*). This analysis suggested a minimal sample size of 13 participants for each group. As such, the sample sizes of our groups (>14) had sufficient power to detect group differences.

For Experiment 1, 15 individuals with CA were tested along with 15 NT healthy participants. For Experiment 2, 27 individuals with CA and 27 NT participants were tested. Three of these CA patients had participated in Experiment 1. Two participants (1/group) were excluded because they had accuracy scores below chance level (50%). For Experiment 3, 22 individuals with CA were tested along with 22 NT participants. Two of these CA patients had participated in Experiment 2. Four participants were excluded (2/group) based on a failure to respond correctly to the attention probes. Thus, in Experiments 1–3, 63 individuals with CA and 63 healthy age- and education-matched NT participants were included in the final analyses: 15/group in Experiment 1, 26/group in Experiment 2, and 20/group in Experiment 3.

The participants' demographic and medical information are presented in *Table 1*. Individuals with a previously established diagnosis of CA were recruited through our clinical Center for Accessible Neuropsychology (CAN) database. The CA groups included participants with genetically confirmed

**Table 1.** Demographic and medical summary table of the cerebellar ataxia (CA) and neurotypical (NT) groups in each experiment. Mean ± SEM.

| Group | Years of education | # Of females | Age | MoCA | SARA |
|---|---|---|---|---|---|
| Experiment 1 | | | | | |
| NT | 16.6±0.8 | 10 | 58.0±1.70 | | |
| CA | 16±0.7 | 13 | 60.6±3.12 | 27.9±0.47 | 13.6±1.4 |
| Experiment 2 | | | | | |
| NT | 16.7±0.8 | 11 | 57.4±2.62 | 27.1±0.78 | NA |
| CA | 16.8±0.7 | 12 | 57.1±2.31 | 26.4±0.81 | 12.2±1.4 |
| Experiment 3 | | | | | |
| NT | 16.1±0.7 | 11 | 57.9±2.43 | 27.3±0.84 | NA |
| CA | 15.8±0.6 | 10 | 58.1±2.10 | 26.2±1.36 | 14.2±1.3 |

SCA6, in which the pathology is relatively limited to the cerebellum (*Klockgether et al., 2019*). Based on a prescreening interview, we included participants who were diagnosed with ataxia, had MRI evidence of cerebellar degeneration, and had genetic confirmation of SCA6. Individuals with other neurological conditions (not CA), psychiatric conditions, learning disabilities, and severe visual or auditory impairments were excluded from the study. The NT participants were recruited through the Prolific platform (*Palan and Schitter, 2018*) for Experiment 1, and through our CAN database for Experiments 2 and 3, targeting participants that met the same demographic criteria as the CA participants in terms of age, years of education, gender distribution, and no general cognitive impairment. As previously mentioned, except in Experiment 1 (where MoCA was not collected), MoCA, age, and years of education were included as covariates in the primary analyses of all the experiments. All participants were above the age of 18 and were required to be able to understand and provide informed consent. The Tel Aviv University Institutional Review Board approved the protocol.

## Neurological and neuropsychological assessment

We followed the online neuropsychological testing published protocol (*Saban and Ivry, 2021a*; *King et al., 2024*; *Gilad et al., 2025*; *Binoy et al., 2023*; *Picciotto et al., 2024*; *Schönfeldová et al., 2025*; *Algon et al., 2025*). Individuals were invited by email to participate in an online, live interview with an experimenter. After providing informed consent, the participant completed a demographic and medical questionnaire. The trained experimenter then administered the Montreal Cognitive Assessment test (MoCA) (*Gilad et al., 2025*; *Binoy et al., 2023*; *Picciotto et al., 2024*; *Schönfeldová et al., 2025*; *Nasreddine et al., 2005*) as a brief evaluation of cognitive status. The CA participants continued to the medical evaluation phase. First, the experimenter obtained the participant's medical history, collecting information about age at diagnosis, medication, primary symptoms, genetic subtype (based on molecular genetic testing), diet, other neurological or psychiatric conditions, and more relevant information. Second, the experimenter administered the Scale for Assessment and Rating of Ataxia (SARA) (*Saban and Ivry, 2021a*; *Picciotto et al., 2024*; *Algon et al., 2025*; *Schmitz-Hübsch et al., 2006*) as an evaluation of disease severity. The mean duration since diagnosis was 7 years in Experiment 1, 6.4 years in Experiment 2, and 6.8 years in Experiment 3. To avoid participants with general cognitive impairments, all CA participants had early to moderate ataxia severity (see SARA scores), providing further evidence for the absence of general cognitive impairments. This session took 40–60 min to complete.

## Procedure

The experiments were programmed in Gorilla Experiment Builder (*Anwyl-Irvine et al., 2020*) and designed to be compatible with personal computers. Stimuli were presented at the center of the screen as black characters on a white background. The actual size in terms of visual angle varied given that participants used their computer system, but we chose a font (7 HTML) that is clearly readable on all screens (as determined by pilots when developing the tasks). Prospective participants were invited by email to participate in an experiment. The email provided an overview of the experimental task and included a link that could be clicked to initiate the experimental session. The link was associated with a unique participant ID, providing a means to ensure that the data was stored in an anonymized and confidential manner. Once activated, the link connected to the Gorilla platform was used to run the experimental session. The instructions were provided on the monitor in an automated manner, with the program advancing under the participant's control.

To maintain and check attentiveness, in each of the three experiments, we included three attention probes that appeared before, during, and after the experimental block (e.g. 'Do not press the Z key to continue; press the M key to continue'). At the end of the task, the participants were asked to provide feedback on their experience (e.g. 'How well were the study instructions explained?').

### Experiment 1 – Cerebellar contribution to symbolic subtraction

First, we manipulated the expectancy effect (expected vs. unexpected) of the equation. While at correct equations (e.g. 9–5–1=3), the presented solution (e.g. 3) is in accordance with the participant's expectation, at incorrect equations (e.g. 9–5–1=4), the presented solution (e.g. 4) violates the participant's expectations (see also *Berger et al., 2006*). Accordingly, RTs are typically higher for incorrect problems compared to correct problems (*Ashcraft, 1992*; *Cohen-Kadosh and Dowker, 2015*; *Saban*

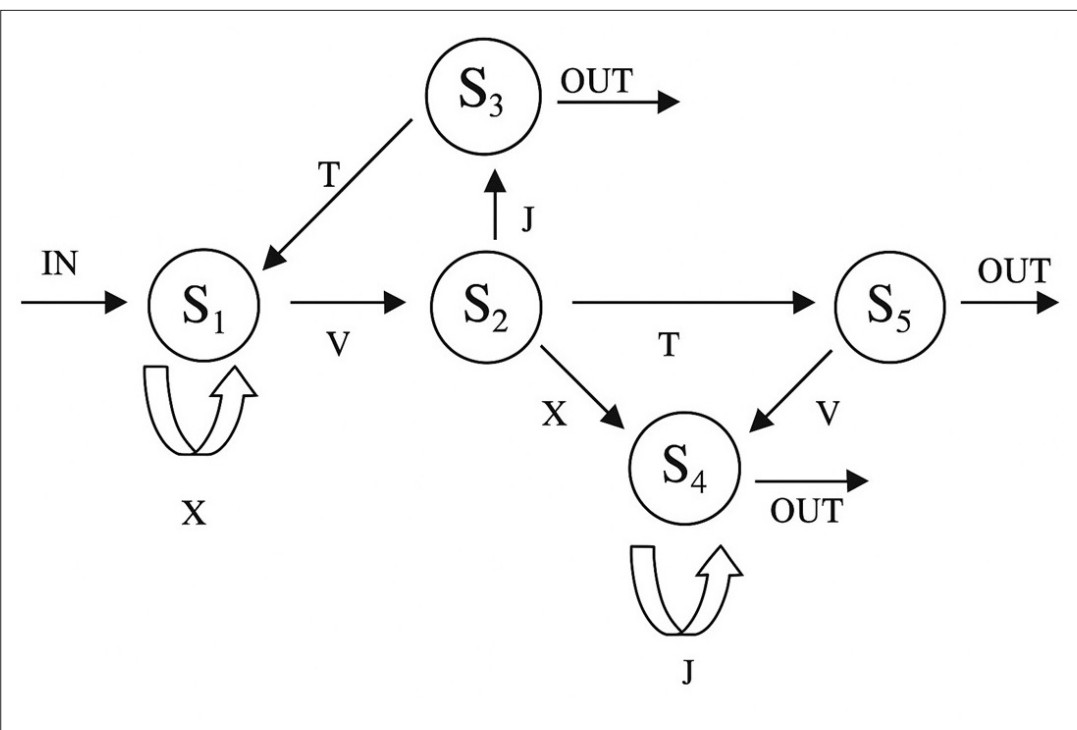

**Figure 6.** The finite-state Markovian rule system used in Experiment 3. S1-S5 indicate the states that occur during the generation of grammatical letter strings. Letter strings are formed by starting at one entry and then by navigating from one transition to another, with each transition being able to generate a letter. A letter string terminates when an exit is reached.

et al., 2021c). We computed the 'expectancy effect' score by subtracting the mean RT of expected equations from the mean RT of unexpected equations. Second, we manipulated the complexity of the equation by probing the number of steps required to solve it. We employed problems that involved either one or two operators. Accordingly, subtracting one single-digit number (e.g. 9–5=4) takes less time than subtracting two single-digit numbers (9–3–2=4) because the latter requires more cognitive steps to solve the problem (*Saban et al., 2024*). We computed the 'complexity effect' score by subtracting the mean RT of low-complexity trials from that of high-complexity trials.

At the beginning of each trial, a fixation cross appeared in the center of the screen for 1000 ms. Then, the fixation cross was replaced by a display of an equation. The participant was required to determine whether a given subtraction equation was correct (by pressing the 'Z' key) or incorrect (by pressing the 'M' key). Three practice equations were presented before the experimental trials to familiarize participants with the procedure. The equation remained on the screen until a response was recorded or until 5 s had elapsed, whichever occurred first. Participants were instructed to respond as quickly and accurately as possible. Visual feedback was provided for 500 ms above the equation, with a green checkmark (√) or a red X indicating the accuracy detected.

In Experiment 1, we created 64 subtraction equations. To minimize the effect of memory, each equation only appeared once. To manipulate the expectancy effect, half (32) of the equations were correct, and half were incorrect. To manipulate complexity, half of the equations were of low complexity level, and the other half were of high complexity level. All the experimental conditions were counterbalanced and presented in a random order. In the middle of the task, the participant had a ten-second break. The experiment lasted approximately 15 min.

## Experiment 2 – Cerebellar contribution to alphabet transformation
Participants completed an alphabet letter transformation task. In this task, participants were presented with alphabet letters transformation based on an arithmetic operator (see *Figure 3*). On the top row, we presented letters (e.g. 'C M') that needed to be transformed by a given rule in a discrete manner for each letter. In the middle row, the transformation rule appeared (i.e. +1 or +2). On the bottom row,

the transformation results appeared (e.g. 'D N'). Participants were required to determine the correctness of the transformation displayed on the screen. Six practice problems were administered before the task to familiarize participants with the necessary procedure. At the onset of each experimental trial, a fixation cross appeared in the center of the screen for 1 s'. Then, the fixation cross was replaced by an alphabet transformation problem. The stimulus remained on the screen until a response was recorded or until 10 s' had elapsed, whichever occurred first. Participants were instructed to respond as quickly and accurately as possible. Visual feedback was presented for 1 s' above the equation, with a green checkmark (√) or a red X indicating the accuracy of the response. If a response was not detected within 9 s', participants received the feedback message 'Respond faster.' RT was calculated on the participants' correct responses.

In Experiment 2, we created 96 unique letter transformation problems. The letters were chosen randomly, but each letter appeared only once in each problem. Similarly to Experiment 1, to minimize the effect of memory, each problem appeared only once. We manipulated the expectancy effect by presenting expected or unexpected alphabet transformation problems (50% of the trials). We also manipulated the level of complexity: The problems either required transforming one letter (low complexity condition) or transforming two letters (high complexity condition). We calculated the complexity effect (complex minus simple problems). All the experimental conditions were counterbalanced and presented in a random order. The experiment lasted approximately 25 min.

## Experiment 3 – Cerebellar contribution to cognitive expectations under uncertainty

Participants were first exposed to strings of letters generated according to a specific novel grammatical rule (*Figure 6*). During this training phase, they were not informed about the rules regarding how to organize a sequence of letters in a specific order. Next, they were shown new strings and asked to judge whether these strings followed the grammatical rules or not. We assessed learning by measuring how well participants could distinguish between grammatical and nongrammatical strings.

During the test stage, we assessed the participant's response to unexpected (nongrammatical) versus expected (grammatical) problems. As there are no expectations based on previous lifetime knowledge, the participant's expectations are formed during the task rather than before it. This approach allowed us to minimize the influence of previously learned top-down processes on newly learned grammar regarding a specific order of a sequence of letters.

We achieved this by using an AGL task (*Chang and Knowlton, 2004*). In this task, during the training phase, we exposed participants to 23 strings of 2–6 letters (e.g. 'XVJ') four times each (92 training trials). In each trial, a fixation cross appeared for 500 ms, and participants were requested to type in the string (each string appeared for 3 s). Next, in the test phase, participants were informed that the order of the letters in the previous strings was determined by a complex set of grammatical rules, without explicitly stating the rules. See *Figure 5* for the Markovian grammar chain used to produce the training and test strings. We then presented novel strings of letters (32 strings, each three times), and the participant was required to decide whether each new string was formed according to the grammatical rule or not. If the new sequence of letters followed the rule, it was considered 'grammatical' (e.g. 'XVXJ'). Otherwise, it was considered 'nongrammatical' (e.g. 'XVXT'). Given the results of the previous two experiments, complexity was not a factor of interest in this experimental design.

Notably, utilizing the AGL task, we also probed novel cognitive expectations under uncertainty. In the test phase, to assess the effect of uncertainty, we manipulated the level of similarity between grammatical and nongrammatical strings. In the low similarity condition, it was relatively easy for participants to distinguish between grammatical and nongrammatical strings. Participants were able, with a higher level of sensitivity, to differentiate which problems adhered to the expected grammar and which deviated from it (i.e. incorrect). In contrast, in the high similarity condition, participants' sensitivity was decreased. The discrimination between grammatical and nongrammatical strings was more challenging given the elevated degree of similarity between them. This manipulation also allowed us to disentangle between the difficulty in learning the rule during training and the difficulty in responding to the new strings during testing.

We presented strings that were utilized in previous studies (*Chang and Knowlton, 2004*). For each test string, we calculated a similarity value, which is sometimes called a 'chunk strength' value. Similarity refers to the frequency of specific letter combinations (bigrams and trigrams) that participants

have been exposed to during the training phase. Higher similarity indicates that certain letter combinations have been repeated more frequently during the training phase. This similarity value served as a measure of uncertainty since, in higher similarity levels, the sensitivity to grammatical versus nongrammatical is lower. Similar to previous studies (*Chang and Knowlton, 2004*), we then categorized the test strings into four groups based on grammaticality and low- or high-similarity values: Grammatical high, grammatical low, nongrammatical high, and nongrammatical low. We have utilized a 2×2 orthogonal design (grammaticality × similarity).

Of the grammatical and nongrammatical test strings (*Chang and Knowlton, 2004*), there were an equal number of high-similarity (i.e. chunk strengths) and low-similarity items. The similarity was calculated as the average number of times each of the bigrams and trigrams in the string had been presented in the training set. The average similarity of high-similarity items was 8.5; the average of low-similarity items was 5.6.

To describe the magnitude of the sensitivity to grammatical status, we calculated a percent correct score for each participant in each condition. The analysis of accuracy not only provided insights into cognitive performance but also served a secondary benefit by reducing potential motor effects on RT. This approach helps to disentangle cognitive processes from motor execution, which is impacted by CA. Typically, in experiments using the AGL task (*Chang and Knowlton, 2004*), the NT group is more accurate in grammatical strings compared to nongrammatical strings. In addition, this group demonstrates higher accuracy in low similarity conditions, where grammatical strings are likely more distinguishable from non-grammatical strings, compared to high similarity conditions. Accordingly, and to control potential response bias/motor-related abilities, a common dependent measure is $d'$ (sensitivity) in discriminating between grammatical vs. nongrammatical strings. To investigate the expectancy effect under uncertainty conditions, our focus in Experiment 3 was on the comparison of the $d'$ between groups in each level of similarity.

## Acknowledgements

This work was supported by Tel Aviv University's new researchers grant for WS.

## Additional information

### Funding

| Funder | Grant reference number | Author |
| --- | --- | --- |
| Tel Aviv University | | William Saban |

The funders had no role in study design, data collection and interpretation, or the decision to submit the work for publication.

### Author contributions

Leonardo Daniel, Formal analysis, Methodology, Writing – original draft, Project administration, Writing – review and editing; Eli Vakil, Writing – review and editing; William Saban, Conceptualization, Resources, Formal analysis, Supervision, Funding acquisition, Validation, Investigation, Visualization, Methodology, Writing – original draft, Writing – review and editing

### Author ORCIDs

William Saban ⓘD https://orcid.org/0000-0003-4215-2478

### Ethics

Human subjects: This study was approved by the Tel Aviv University (TAU) Institutional Review Board-Committee (#0005713-4). Informed consent was obtained from all individual participants included in the study.

Reviewer #1 (Public review): https://doi.org/10.7554/eLife.105864.3.sa1
Author response: https://doi.org/10.7554/eLife.105864.3.sa2

## Additional files

### Supplementary files
MDAR checklist

### Data availability
All data supporting the conclusions of this study are included in the main text. The raw datasets analyzed in this study involve patients with a rare disease (prevalence < 0.01), and publishing these data poses a significant risk to patient privacy. For this reason, the patients' individual raw data cannot be made publicly available. Researchers who wish to access the raw data may contact the corresponding author (willsabanATtauex.tau.ac.il). Requests must include a project proposal outlining the intended use of the data. Proposals will be reviewed by our Institutional Review Board (IRB) and/or data access committee to ensure compliance with ethical and legal standards. If approved, access will be granted for non-commercial research purposes only. The review process typically requires 4-8 weeks. Summary data files and R code to reproduce the figures and statistical results are available on GitHub (copy archived at *Saban, 2025*).

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
