## [Editor Report · eLife Assessment]

This **valuable** investigation provides new and **solid** evidence for a specific cognitive deficit in cerebellar degeneration patients. The authors use three tasks that modulate complexity and violations of cognitive expectations. They show specific slowing of reaction times in the presence of violations but not with task complexity. While some alternative interpretations of the results are possible and are discussed, the work provides a new, invaluable data point in describing the cognitive contribution of cerebellar processing.

---

## [Referee Report · Reviewer #1 (Public review)]

Summary:

The authors test the hypothesis that the contribution of the cerebellum to cognitive tasks is similar to motor tasks, and is related to the processing of prediction errors (here: violation of expectations, VE). In three experiments, they find that cerebellar patients show differences compared to controls in measures of VE, but not task complexity. The findings show that cerebellar disease results in deficits in VE processing in cognitive tasks, and makes a valuable contribution of the field. The authors were able to test a large number of patients with cerebellar disease which is known to primarily affect the cerebellum (i.e. SCA6).

Strengths:

A strength of the study is that it is hypothesis-driven and that the three experiments are very well thought out. Furthermore, a comparatively large group of patients with spinocerebellar ataxia type 6 (SCA6) was tested, a disease which affects primarily the cerebellum.

Weaknesses:

- Acquisition of brain MRI scans would have been useful to perform lesion-behaviour-mapping. But this does not limit the significance of the behavioural findings.

- Exp. 1 and 2: The lack of difference in accuracy was that an unexpected finding? How meaningful are the used paradigms when accuracy was the same in cerebellar patients and controls?

- Exp. 1 and 2: Cerebellar patients have motor dysfunction which impacts reaction time. Can the authors exclude that this contributed at least in part to their findings? Any correlations to SARA score (upper limb function) or oculomotor dysfunction (e.g. presence of nystagmus)?

- Data on the attention probes which have been done would be of interest. Were there any differences in attention between patients and controls, any correlations with the findings?

Comments on revisions:

I am not sure if I can follow the interpretation of the authors that the cerebellum contributes to prediction errors, but not predictions; These two are tightly connected? It may rather be that in patients with slowly progressive chronic disease there is a lot of compensation? It is not so rare that in cognitive tasks cerebellar patients do not perform differently from controls, even though one would expect a difference (e.g. based on fMRI data in controls)? Another factor which likely adds is age, Patients and controls are often middle-aged and elderly, adding to variability, decreasing the chance to see group differences?

---

## [Author Response]

The following is the authors’ response to the original reviews

**Joint Public Review:**
Summary:In this study, Daniel et al. used three cognitive tasks to investigate behavioral signatures of cerebellar degeneration. In the first two tasks, the authors found that if an equation was incorrect, reaction times slowed significantly more for cerebellar patients than for healthy controls. In comparison, the slowing in the reaction times when the task required more operations was comparable to normal controls. In the third task, the authors show increased errors in cerebellar patients when they had to judge whether a letter string corresponded to an artificial grammar.Strengths:Overall, the work is methodologically sound and the manuscript well written. The data do show some evidence for specific cognitive deficits in cerebellar degeneration patients.

Thank you for the thoughtful summary and constructive feedback. We are pleased that the methodological rigor and clarity of the manuscript were appreciated, and that the data were recognized as providing meaningful evidence regarding cognitive deficits in cerebellar degeneration.

Weaknesses:The current version has some weaknesses in the visual presentation of results. Overall, the study lacks a more precise discussion on how the patterns of deficits relate to the hypothesized cerebellar function. The reviewers and the editor agreed that the data are interesting and point to a specific cognitive deficit in cerebellar patients. However, in the discussion, we were somewhat confused about the interpretation of the result: If the cerebellum (as proposed in the introduction) is involved in forming expectations in a cognitive task, should they not show problems both in the expected (1+3 = 4) and unexpected (1+3=2) conditions? Without having formed the correct expectation, how can you correctly say "yes" in the expected condition? No increase in error rate is observed - just slowing in the unexpected condition. But this increase in error rate was not observed. If the patients make up for the lack of prediction by using some other strategy, why are they only slowing in the unexpected case? If the cerebellum is NOT involved in making the prediction, but only involved in detecting the mismatch between predicted and real outcome, why would the patients not show specifically more errors in the unexpected condition?

Thank you for asking these important questions and initiating an interesting discussion. While decision errors and processing efficiency are not fully orthogonal and are likely related, they are not necessarily the same internal construct. The data from Experiments 1 and 2 suggest impaired processing efficiency rather than increased decision error. Reaction time slowing without increased error rates suggests that the CA group can form expectations but respond more slowly, possibly due to reduced processing efficiency. Thus, this analysis of our data suggests that the cerebellum is not essential for forming expectations, but it plays a critical role in processing their violations.

Relatedly, a few important questions remain open in the literature concerning the cerebellum’s role in expectation-related processes. The first is whether the cerebellum contributes to the formation of expectations or the processing of their violations. In Experiments 1 and 2, the CA group did not show impairments in the complexity manipulation. Solving these problems requires the formation of expectations during the reasoning process. Given the intact performance of the CA group, these results suggest that they are not impaired in forming expectations. However, in both Experiments 1 and 2, patients exhibited selective impairments in solving incorrect problems compared to correct problems. Since expectation formation is required in both conditions, but only incorrect problems involve a VE, we hypothesize that the cerebellum is involved in VE processes. We suggest that the CA group can form expectations in familiar tasks, but are impaired in processing unexpected compared to expected outcomes. This supports the notion that the cerebellum contributes to VE, rather than to forming expectations.

In Experiment 3, during training, the participant is learning a novel rule (grammar), forming new expectations on how strings of letters should be. Afterwards, during testing, the participant is requested to identify if a novel string is following the rule or not. We examined sensitivity to distinguish between grammatical and non‐grammatical strings of letters, thus taking into account a baseline ability to identify expected strings. Additionally, both in the low‐similarity and highsimilarity conditions, there are expectations regarding whether the strings are following the rule or not. However, in the high‐similarity condition, there is more uncertainty regarding which strings are following the grammatical rule, as demonstrated in a lower sensitivity (d prime). Given the group differences only in the low similarity condition, these results suggest the CA group is impaired only when the rules are more certain. Given these results, we suggest that forming cognitive expectations is not necessarily dependent on the cerebellum. Rather, we propose that the cerebellum is critical for processing rule-based VE (detection or processing of detected errors) under conditions of more certainty. One remaining question for future studies is whether the cerebellum contributes to detection of a mismatch between the expectation and sensory evidence, or the processing of a detected VE.

We suggest that these key questions are relevant to both motor and non-motor domains and were not fully addressed even in the previous, well-studied motor domain. Importantly, while previous experimental manipulations17,19,40,94–96 have provided important insights regarding the cerebellar role in these processes, some may have confounded these internal constructs due to task design limitations (e.g., lack of baseline conditions). Notably, some of these previous studies did not include control conditions, such as correct trials, where there was no VE. In addition, other studies did not include a control measure (e.g., complexity effect), which limits their ability to infer the specific cerebellar role in expectation manipulation.

Thus, the current experimental design used in three different experiments provides a valuable novel experimental perspective, allowing us to distinguish between some, but not all, of the processes involved in the formation of expectations and their violations. For instance, to our knowledge, this is the first study to demonstrate a selective impairment in rule-based VE processing in cerebellar patients across both numerical reasoning and artificial grammar tasks. If feasible, we propose that future studies should disentangle different forms of VE by operationalizing them in experimental tasks in an orthogonal manner. This will allow us to achieve a more detailed and well-defined cerebellar motor and non-motor mechanistic account.

**Recommendations for the authors:**

**Editors comments:**
The Figures are somewhat sub-standard and should be improved before the paper is made the VOR. Ensure consistent ordering of the group factor (CA, NT) and experimental factor across Figure 3,4, and 6 (panels A). Having the patient group as columns in Figure 4a and in rows in Figure 6a is very confusing.

We have standardized the layout across Figures 2, 4, and 6 so that the group factor (CA, NT) and experimental conditions are consistently ordered. In all panels, the group factor now appears as a column.

Subpanels should be numbered A,B,C... not A, B1, B2.

Subpanel labels have been updated to follow the standard A, B, C format across all figures.

Fonts should have a 100% aspect ratio - they should not be stretched (Figure 6B).

We have corrected the font aspect ratios in all figures (e.g., Figure 6B) to ensure proper proportions and readability.

Colors should be more suitable to print - use a CYMK color scheme (i.e. avoid neon colors such as the neon green for the CA).

The color scheme across all figures has been revised to be print-friendly using CMYKcompatible, colorblind-accessible palettes. Neon green for the CA group was replaced with a more muted, distinguishable color.

Abstract: "The CA group exhibited a disproportionate cost when comparing expected problems compared to unexpected problems" - I recommend switching unexpected and expected, as the disproportional cost in on the former.

We have changed the wording of the sentence accordingly.

Upon re-reading the details for the AGL task were not clear to us. Please do not rely on the reference (78) for the details - your paper should contain enough information to have the reader understand the experimental details. For you to appreciate the depth of our not-understanding, here a simple question: The test strings either followed the grammar in Fig 5 or they did not. If they did not, how exactly was similarity to the grammar measured? If they did, what was the difference between the “Grammatical-high” and “Grammatical-low” trials? If the string was grammatical, there should not be a notion of similarity, no? Or where these trials arbitrary split in half?

We have clarified that 50% of the test strings followed the grammar of the training strings. We also elaborated on the calculation of chunk strength as a measure of similarity between the training and testing strings, similar to the previous papers. The differences between low and high similarity are explained in the paper. Specifically, for each test string, we calculated chunk strength by summing the frequencies of all relevant substrings (e.g., bigrams and trigrams) that appeared in the training set. The test strings whose chunk‐strength values fell above the median for grammatical items were classified as “high similarity,” while those falling below the median were classified as “low similarity.” Also, grammatical strings can be of both low and high similarity; this is precisely the beautiful aspect of this experimental manipulation, showing the importance of uncertainty. We have utilized a 2 × 2 fully orthogonal design (grammaticality × similarity).

Experimental details of the task should be added to the Method section. In the results you should only mention the experimental details that are necessary for understanding the experiments, but details such as the number of trials, etc, can be moved to the methods.

We have now moved the experimental task details to the Method sections.

**Reviewer #1 (Recommendations for the author):**
Studies have been done online and not in the lab. Could that have affected the results?

We addressed this in the Methods section, referring to established protocols for online neuropsychological testing[9–12]. Our results align with similar in-lab findings in both the subtraction and AGL tasks, supporting the online approach's robustness.

Figure 2, B1; Figure 4, B1; Figure 6B: How many patients performed worse than the (worst-performing) controls? There appears to be quite some overlap between patients and controls. In the patients who performed worse, was there any difference from the other patients (e.g. disease severity as assessed by SARA score, repeat length, data of attention probes)?

We appreciate the reviewer’s thoughtful comment. We considered conducting individual-level comparisons to identify patients who performed worse than the lowest-performing controls. However, defining "worse" based on the performance of the lowest control is only one possible criterion. Other definitions—such as a specific number (1/2/3?) of standard deviations below the control mean—are also commonly used in literature, and each may yield different conclusions. This variability highlights the lack of a standardized threshold for what constitutes “worse” or "impaired" performance at the individual level. Given this ambiguity, and in line with prior studies that focus on average group differences rather than “impairment” prevalence, we chose not to include these individual-level comparisons. We believe this approach better aligns with the goals and design of the current study. That said, we agree that examining individual variability is important and may be more appropriate in future studies with larger samples so that percentage is a more robust measure. However, given the rarity of the disease, this would also be a challenge for future studies.

SARA ataxia scale does not include oculomotor function. In SCA6 oculomotor deficits are frequent, eg, downbeat nystagmus. Please include information on oculomotor dysfunction.

We thank the reviewer for this important observation. While it is true that the SARA scale does not explicitly assess oculomotor function, our experimental design – in all three experiments – has control conditions that help account for general processing differences, including those that could arise from oculomotor deficits. These conditions, such as the correct trials and the complexity effects, allow us to isolate effects specifically related to the violation of expectation while minimizing the influence of broader performance factors, such as eye movement abnormalities. We also note that, while some patients can experience oculomotor symptoms such as downbeat nystagmus, none of our tasks required precise visual tracking or gaze shifts. In our experimental tasks, stimuli were centrally presented, and no visual tracking or saccadic responses were required. Moreover, the response time windows and stimulus durations (>2–5 s) were sufficient to mitigate the effects of delayed visual processing due to oculomotor impairment.

Why was MoCA used and not the CCAS-Schmahmann scale to assess cognitive function?

We selected the MoCA due to its broad clinical utility, time efficiency, and ability to detect mild cognitive impairment specifically in CA[101,102].

Were there any signs of depression in the patient group that could have affected the results?

None of the patients had a clinical diagnosis of depression or were undergoing psychiatric treatment.

Additionally, the interaction between group and expectancy was insignificant when RT was the depended vaibale .." = variable

This has been corrected to "variable" in the revised manuscript.

**Reviewer #2 (Recommendations for the authors):**
The terms 'unexpected' and 'expected' conditions are confusing. [...] Terming this 'violation of expectation' seems unnecessarily complicated to me.

We thank the reviewer for raising this important concern. We recognize that the terms "expected" and "unexpected" can be ambiguous without clarification, and that "violation of expectation" (VE) may initially appear unnecessary. Our choice to use VE terminology is grounded in an established theoretical framework that distinguishes between mere stimulus correctness and prediction mechanisms. Specifically, VE captures the internal processing of mismatches between anticipated and observed outcomes, which we believe is central to the cerebellar function under investigation. While simpler, technical alternatives (e.g., "correct" vs. "incorrect") could describe the stimuli, we find that VE more accurately reflects the mental constructs under study and is consistent with previous literature in both motor and cognitive domains.

Both tasks provide an error (or violation of expectation) that is non-informative and therefore unlikely to be used to update a forward model. The authors draw on motor literature to formulate a cognitive task where the presence of an error would engage the cerebellum and lead to longer reaction times in cerebellar patients. But in the motor domain, mismatch of sensory feedback and expectations would lead to an updating of the internal forward model. It seems unlikely to me in the arithmetic and alphabetic addition tasks that patients would update their internal model of addition according to an error presented at the end of each trial. If the error processed in these tasks will not lead to the updating of the internal forward model, can the authors discuss to what extent the cerebellum will be engaged similarly in these tasks, and what exactly connects cerebellar processing in these motor and cognitive tasks.

We thank the reviewer for this thoughtful and important comment. We fully agree that the current tasks do not directly probe learning-related updating of internal models. As stated in the paper, the goal of the present study was not to support or refute a specific claim regarding the cerebellum’s role in learning processes. Rather, our focus was on examining cerebellar involvement in the processing of VE. While we were inspired by models from the motor domain, our design was not intended to induce learning or adaptation per se, but to isolate the processing of unexpected outcomes. We agree that the tasks in their current form are unlikely to engage forward model updating in the same way as in sensorimotor adaptation paradigms. That said, we believe the current findings can serve as a basis for future research exploring the relationship between cerebellar prediction error processing and learning over time. As we also noted in the paper, this is a direction we propose, and actively pursuing, in ongoing research work.

The colour scheme is difficult for anyone with colour blindness or red-green visual impairment. Please adjust.

All figures have been revised to use CMYK-compatible, colorblind-safe palettes, and neon colors have been removed.

The introduction is a bit difficult to understand, because the authors draw on a number of different theories about cerebellar functioning, without clearly delineating how these relate to each other. For example: (a) In the paragraph beginning with 'notably': If the cerebellum is required for sequential operations, why does it show the impairment with the rotation of the letters?

We understand the concern that if the cerebellum is involved in sequential operations, its involvement in mental letter rotation, which can be assumed as “continuous transformation,” may appear contradictory. We note that the boundary between continuous and stepwise, procedural operations is not always clear-cut and may vary depending on the participant's strategy or previous knowledge, which is not fully known to the researchers. Furthermore, to our knowledge, prior work on mental rotation has not directly investigated the impact of VE during this task. However, these are two debatable considerations.

More importantly, a careful reading of our paper suggests that our experiments were designed to examine VE within tasks that involve sequential processing. Notably, we are not claiming that the cerebellum is involved in sequential or procedural processing per se. Rather, our findings point to a more specific role for the cerebellum in processing VE that arises during the construction of multistep procedural tasks. In fact, the results indicate that while the cerebellum may not be directly involved in the procedural process itself, it is critical when expectations are violated within such a context. This distinction is made possible in our study by the inclusion of a control condition (the complexity effect), which allows for a unique dissociation in our experimental design—one that, to our knowledge, has not been sufficiently addressed in previous studies.

Additionally, in the case of arithmetic problem solving—such as the tasks used in prior studies cited in our manuscript21—there is substantial evidence that these problems are typically solved through stepwise, procedural operations. Arithmetic reasoning, used in Experiments 1 and 2, has been robustly associated with procedural, multi-step strategies, which may be more clearly aligned with traditional views of cerebellar involvement in sequential operations. Thus, we propose that the role of the cerebellum in continuous transformations should be further examined.

We suggest a more parsimonious theory that the cerebellum contributes to VE, a field that was highly examined before. Yet, to reconcile ours and previous findings, we propose that the cerebellum’s contribution may not be limited to either continuous or stepwise operations per se, but rather to a domain-general process: the processing of VE. This theoretical framework can explain performance patterns across both mental rotation tasks and stepwise, procedural arithmetic.

The authors mention generation prediction as a function of the cerebellum, processing of prediction errors (or violations of expectations), sequentially, and continuous transformations - but it is unclear whether the authors are trying to dissociate these from each other or whether ALL of these functions have informed task design.

We propose that the cerebellum’s contribution may not be limited to either continuous transformations or stepwise, procedural operations per se, but rather to a domain-general process: the processing of VE. We would like to clarify that we do not claim the cerebellum contributes to continuous transformations only, as suggested in some earlier work[21]. Rather, it could be that the cerebellum may contribute to continuous transformations, but we propose that it also supports multi-step, procedural processes. Given that framework, in the current study, across three separate experiments, we demonstrated that the cerebellum can also contribute to procedural, multi-step reasoning tasks.

Minor CommentsTypo under paragraph beginning with 'notably' - cerebellum role should be cerebellar role.

Corrected as suggested.

When mentioning sequences as a recruiting feature for the cerebellum in the introduction, Van Overwalle's extensive work in the social domain should be referenced for completeness.

Thank you for the suggestion. We have now cited Van Overwalle’s work on cerebellar involvement in sequence processing within the social domain in the revised Introduction.